# Nucleoside Diphosphate Kinases Are ATP-Regulated Carriers of Short-Chain Acyl-CoAs

**DOI:** 10.3390/ijms25147528

**Published:** 2024-07-09

**Authors:** Domenico Iuso, Julie Guilliaumet, Uwe Schlattner, Saadi Khochbin

**Affiliations:** 1University of Teramo, Department of Veterinary Medicine, 64100 Teramo, Italy; 2University Grenoble-Alpes, CNRS UMR 5309, INSERM U1209, Institute for Advanced Biosciences, 38706 La Tronche, France; julie.guilliaumet@univ-grenoble-alpes.fr (J.G.); saadi.khochbin@univ-grenoble-alpes.fr (S.K.); 3University Grenoble-Alpes, INSERM U1055, Laboratory of Fundamental and Applied Bioenergetics (LBFA), 38058 Grenoble, France; uwe.schlattner@univ-grenoble-alpes.fr; 4Institut Universitaire de France, 75231 Paris, France

**Keywords:** nucleoside diphosphate (NDP) kinases 1 and 2 (NME1/2), acyl-CoAs, fatty acid synthesis, histone acetylation

## Abstract

Nucleoside diphosphate (NDP) kinases 1 and 2 (NME1/2) are well-characterized enzymes known for their NDP kinase activity. Recently, these enzymes have been shown by independent studies to bind coenzyme A (CoA) or acyl-CoA. These findings suggest a hitherto unknown role for NME1/2 in the regulation of CoA/acyl-CoA-dependent metabolic pathways, in tight correlation with the cellular NTP/NDP ratio. Accordingly, the regulation of NME1/2 functions by CoA/acyl-CoA binding has been described, and additionally, NME1/2 have been shown to control the cellular pathways consuming acetyl-CoA, such as histone acetylation and fatty acid synthesis. NME1/2-controlled histone acetylation in turn mediates an important transcriptional response to metabolic changes, such as those induced following a high-fat diet (HFD). This review discusses the CoA/acyl-CoA-dependent NME1/2 activities and proposes that these enzymes be considered as the first identified carriers of CoA/short-chain acyl-CoAs.

## 1. Introduction

NME1 and NME2 are members of the large conserved NME protein family, both localizing to the nucleo-cytosolic cellular compartment. Given their high degree of structural and functional similarity, they are collectively referred to as NME1/2 in this review. Their function was long considered to be restricted to their nucleoside diphosphate kinase (NDPK) activity, maintaining cellular nucleoside triphosphate (NTP) homeostasis, in particular of NTPs other than ATP. Therefore, in principle, the NME1/2 NDPK activity can control processes such as DNA/RNA-synthesis or GTP-dependent biological activities, and hence mediate basic cellular functions such as proliferation, differentiation, development, and metastasis of cancer cells. In recent years, NME proteins emerged as true multi-functional proteins. For example, there is evidence that NME1/2 could also act as histidine kinases transferring a phosphate group from its catalytic histidine onto a histidine on other proteins [1,2]. There are many excellent reviews on these known properties of NME1/2 [3], which will not be covered here. Remarkably, during the last two years, three groups independently published the discovery of coenzyme A (CoA) and acyl-CoA binding by NME1/2 [4,5,6,7]. Two of these reports discovered CoA-binding activity of NME1/2 following the capture of cellular proteins in cellular or nuclear protein extracts by CoA-covered beads [4,5]. The third group used an acyl-CoA with a C14-long acyl chain for the capture experiment [6]. While all three approaches demonstrated the capture of NME1/2 from the extracts used, the first two studies further showed binding of NME1/2 to CoA as well as to short-chain acyl-CoAs (SCA-CoAs) [4,5], and approached the functional significance of this interaction. These studies have therefore opened new avenues to yet unknown functions of NME proteins. Our own data suggest that NME1/2 could be considered as cellular CoA/SCA-CoA transporters. The existence of such CoA/SCA-CoA carriers is a new concept, since until now these molecules were thought to diffuse freely within cells. This is in contrast to long-chain acyl-CoAs, for which a well-known transporter, acyl-CoA-binding protein (ACBP), has been structurally and functionally characterized [8].

This review summarizes the major findings regarding this newly discovered activity of NME1/2, and examines the putative cellular functions that rely on CoA and acyl-CoA binding of NME1/2.

## 2. CoA/SCA-CoA-Binding by NME1/2 Could Be Regulated by the Cellular NTP/NDP Ratio

X-ray structural studies of NME1 with different substrates [4,6,7], along with biophysical and biochemical analyses of enzyme–substrate interactions with NME1/2, have unveiled competitive binding between CoA/SCA-CoA and ATP/ADP within the NME active site. This competition occurs independently at each monomer within the NME1 hexameric structure, resulting in the occupancy of up to six sites per hexamer by either ligand. Determination of binding affinities revealed that NME1 binds CoA species at least as well or even better than the canonical NDP substrate. Importantly, other metabolites carrying adenylate moieties, such as NAD, FAD, and ADP-ribose, do not show interaction with NME1 [4].

Binding of CoA/SCA-CoA and ATP/ADP occurs within the known nucleotide binding pocket of NME1 and involves in both cases the common adenylate moiety. However, despite the apparent chemical similarity, their binding modes are completely different, with each engaging a partially distinct set of residues [4]. Important differences arise at the catalytic His118. Typically, His118 interacts with the ATP γ-phosphate and becomes phosphorylated as part of the sequential NDPK reaction. However, with bound CoA, His118 interacts with the 3′-phosphate of the adenosine ribose, only present in CoA, while the α- and β-phosphates of CoA protrude out of the binding pocket. Consequently, phosphorylated His118 obstructs the binding of CoA and acyl-CoAs. Furthermore, since the CoA pantetheine tail and any connected acyl chain lack specific interactions with NME1, the protein is rather non-specific with respect to the SCA-CoA (or acyl-CoA) species. In summary, specific interactions and unique binding determinants of NME1 determinate the binding of either ATP/ADP or CoA/SCA-CoA.

This renders NME1/2 a rare case of an enzyme’s active site capable of binding two different kinds of ligands for two independent functions. The binding of NTPs (and dNTPs, but physiologically mainly ATP) will trigger the NDPK enzymatic function, which transiently phosphorylates NME1/2 at the catalytic His118 residue before this phosphate is transferred to any NDP (or dNDP) subsequently binding to this site. In the case of CoA/SCA-CoA, the function rather consists of reversibly binding the ligands, thus transiently removing them from the soluble pool, i.e., properties akin to those of a carrier or storage protein as discussed further below.

While the overall occupancy of the NME1/2 active sites by either NTP/NDP or CoA/SCA-CoA is predominantly influenced by the local concentrations of these metabolites, there is also an additional regulatory aspect based on the divergent effects of ATP (NTP) and ADP (NDP) binding. The binding of NTPs not only competitively releases bound CoA/SCA-CoA but also phosphorylates His118, thereby stably preventing the rebinding of CoA/SCA-CoA. Conversely, NDP-dependent dephosphorylation of His118 promotes CoA/SCA-CoA binding (see Figure 1). Consequently, high local NTP concentrations will favor the soluble pool of CoA/SCA-CoA, whereas high local NDP concentrations will favor NME1/2-bound CoA/SCA-CoA. Considering NME proteins as carriers of CoA/SCA-CoA, their occupancy will be sensitive to the local NTP:NDP ratio, which has significant functional implications for the utilization of CoA/SCA-CoA depending on cellular energy levels (see Figure 1).

## 3. NME1/2 as ATP-Regulated Short-Chain Acyl-CoA Carriers: A New Regulatory Layer in Energy Homeostasis

SCA-CoAs are generated along various metabolic pathways, mostly glucose oxidation, ß-oxidation, and amino acid catabolism leading to the production of acetyl-CoA. The intermediate SCA-CoAs predominantly generated along these metabolic pathways in mitochondria can be exported via different, partially unknown pathways to the nucleo-cytosolic compartment. There, they are used, e.g., for protein modification by lysine acylations or lipid biosynthesis from acetyl-CoA [9]. These acyl-CoAs represent the activated form of fatty acids and are thus intrinsically highly reactive; they can non-enzymatically modify proteins, including histones, albeit at low rate [10,11,12]. Therefore, it is of vital importance to protect cells of unregulated, non-enzymatic protein modification that could be deleterious [13]. One way to prevent undesirable protein modification is to sequester and carry these reactive acyl-CoA molecules by specific carriers. Such carriers of SCA-CoAs would prevent the free distribution of reactive acyl-CoAs in various cellular compartments and their non-controlled decay following random transfer of their acyl moieties to protein lysine residues. However, no carrier of SCA-CoAs has been identified so far. Our findings strongly suggest that NME1/2, by binding and carrying SCA-CoAs, could protect SCA-CoAs until their use by specific enzymes such as acetyl-CoA carboxylase 1 (ACC1) or different histone acyl transferases (HATs) [4] (Figure 2).

In addition, SCA-CoA binding to NME1/2 could introduce a powerful and yet unknown regulatory level by limiting the access of SCA-CoA to the consuming enzymes. Indeed, we have found an inhibitory role of NME1/2 on de novo lipogenesis (DNL) during high-fat diet (HFD) feeding. The absence of NME2 in liver enhances the accumulation of fatty acid (FA) species, suggesting an enhancement of DNL. We speculate that NME1/2 sequestration of acetyl-CoA limits its use by ACC1 and down-regulates DNL at its first step. In the absence of NME2, ACC1 would have increased access to acetyl-CoA, leading to the observed increase in the concentration of various FA species.

This scenario also allows us to propose a novel level of energy sensing and energy storage. As discussed above, our structural investigations showed that the catalytic histidine phosphorylation of NME1/2 is incompatible with SCA-CoA binding [4]. Thus, in the presence of excess free energy, high ATP levels and high ATP:ADP ratios would prevent acetyl-CoA binding by NME1/2 and hence could enhance its use by ACC1/2. An increased generation of malonyl-CoA by ACC1/2 would then inhibit mitochondrial FA import and ß-oxidation, and at the same time accelerate DNL and the accumulation of triglycerides [14]. This completely new mechanism would come in addition to the well-known ACC1 inhibition by AMP-activated protein kinase (AMPK), occurring under limiting free energy, which leads to increased ß-oxidation and repressed DNL [15].

The same ATP-regulated mechanism could also control protein/histone acetylation. Indeed, high ATP levels would make acetyl-CoA abundantly available to HATs. This could lead to massive histone acetylation (see below). In addition to the use of acetyl-CoA for carbon storage in the form of triglycerides by DNL, histone acetylation may also represent a mechanism to store acetate, accessible when necessary via an acetate-generating system such as histone deacetylase (HDACs) [16] (Figure 2). Collectively, the acetyl-CoA carrier activity of NME1/2 could be considered as a new layer of energy sensing and carbon storage in the cells.

## 4. Comparing Acyl-CoA-Binding Proteins: NME 1/2 versus ACBPs/DBI

Our initial interest in the global histone acetylation-dependent genome reprogramming in mouse spermatogenic cells during their post-meiotic maturation led to the identification of NME 1/2 and DBil5 (acyl-CoA-binding protein/DBI testis-specific protein) [4] as the primary CoA-binding proteins in the testis. We then also identified NME1/2 as the predominant CoA-binding proteins in the liver. However, the liver is known to expresses multiple acyl-CoA-binding proteins (ACBPs), albeit in smaller quantities compared to NME1/2 [4]. While it is established that ACBPs/DBIs serve as carriers of medium-/long-chain acyl-CoAs [17], before our study, the existence of a carrier for CoA and short-chain acyl-CoA remained elusive. Both NME1/2 and ACBPs primarily bind the CoA moiety of acyl-CoAs, with the 3’ phosphate of the CoA ribose playing a crucial role in the fixation and stabilization of CoA [4,18].

ACBPs distinctively engage in an interaction with the acyl chain (e.g., of palmitoyl-CoA), situated between the hydrophobic surfaces of CoA and the protein [18]. Conversely, the structure of NME1/2 complexed with acetyl-CoA, and succinyl-CoA did not reveal any interaction between the protein and the acyl moiety of these molecules [4]. Finally, compared to ACBPs, NME1/2 stands out due to its NDPK activity that involves an NME histidine phosphorylation mediated mainly by ATP, and which affects CoA binding [4], as discussed above.

Based on our findings, we propose that NME1/2 and ACBPs may operate upstream of CoA/short-chain and long-chain acyl-CoAs, respectively, influencing metabolic pathways involving these molecules [19].

In summary, the levels of various acyl-CoAs at steady state are influenced by nutrient availability and cell metabolism. However, their accessibility for specific metabolic pathways is governed by two families of proteins, namely NME1/2 and ACBPs, which demonstrate differing affinities towards acyl-CoAs. The cellular energy status, particularly the pool of free local ATP, additionally regulates acyl-CoA accessibility through competitive binding and phosphorylation by NME1/2. Therefore, we propose a dynamic interplay between acetyl-/acyl-CoA availability and cellular energy status, with Nme1/2 identified as an energy sensor, a function distinct from that of ACBP/DBI proteins.

## 5. NME1/2-Mediated Control of Histone Acetylation

HFD triggers a transcriptional response in mouse liver cells, involving the repression or activation of a specific set of genes [20]. This change in gene expression is associated with the suppression of DNL and activation of a liver protective response (i.e., genes that are normally activated following liver injury and regeneration) [4,21,22]. Our investigations with NME2 knock-out mice revealed that NME2 is a major player in the reprogramming of HFD-dependent gene expression [4].

With these findings, the question arises as to how NME1/2 can control a specific gene expression program, specifically in the context of HFD. An answer to this question could also be based on the acetyl-CoA carrier activity of NME1/2. Our data indicated that under HFD, NME2 mediates a local increase in histone H3K9 acetylation, without any noticeable change in the total chromatin H3K9 acetylation level. Since this specific gain of H3K9 acetylation is strictly dependent on NME2, we can hypothesize that one of the functions of NME2 is the control of targeted histone acetylation, in response to environmental challenges such as an HFD.

The redistribution of histone acetylation has already been reported in the quiescence exit in Saccharomyces cerevisiae. Indeed, a previously published work elegantly demonstrated that under quiescence exit, acetyl-CoA synthetase, Acs2 (the ortholog of mammalian ACSS2), is recruited on active genes’ chromatin. The proposed model is that acetate released following HDAC activity and the CoA released following HAT activities are used by Acs2 to locally regenerate acetyl-CoA and then lead to a targeted histone acetylation at sites where Acs2 is recruited [23]. Another example, which could be more relevant to our discussion here, is the redistribution of histone acetylation under carbon starvation and the subsequent restriction of the total cellular acetyl-CoA concentration, also observed in Saccharomyces cerevisiae. Under these conditions, growth-associated genes lose their H3K9 acetyl group, which can be used to regenerate acetyl-CoA for a targeted H3K9 acetylation on other genes involved in gluconeogenesis and fat metabolism [24]. These phenomena recall our observations in liver cells, where H3K9 acetylation increases on a specific set of genes that show enhanced transcriptional activity under the HFD. However, in our case, we know that NME2 is absolutely required to observe this specific gain of H3K9 acetylation.

This leads to the next question: what is the molecular basis of this activity of NME2? The most probable scenario could involve the ATP-regulated acetyl-CoA carrier activity of NME1/2. Indeed, the HDAC/HAT/ACSS2-dependent regeneration of nuclear acetyl-CoA described above could be captured by NME1/2 and be released in the vicinity of chromatin regions exposed to a high concentration of ATP. The question of localized nuclear production of ATP has received attention in the work of Beato’s laboratory, revealing that the pyrophosphatase NUDIX5 is capable of producing ATP locally from poly-ADP-ribose [25]. Such a local source of ATP could release acetyl-CoA from the NME1/2 carriers at active chromatin regions.

As already outlined above, our structural and functional data show that ATP binding by NME1/2 and subsequent catalytic histidine phosphorylation could increase the pool of unbound acetyl-CoA [4]. Therefore, at chromatin regions undergoing remodeling due to active ATP-dependent remodelers and the presence of local ATP (e.g., generated by NUDIX5), the local ATP could lead to a release of acetyl-CoA bound to NME1/2. This locally released acetyl-CoA could then be used to ensure the observed targeted histone acetylation. The source of this NME1/2-bound acetyl-CoA may also originate from the cytosol. This is because an HFD inhibits DNL, which is a major consumer of cytosolic acetyl-CoA, making it more likely to be available for NME1/2 interaction.

We designed an experiment to prove in vitro the concept of ATP-dependent regulation of acetyl-CoA availability for histone acetylation. Indeed, the addition of acetyl-CoA and an equimolar amount of NME1 resulted in the masking of acetyl-CoA from HATs, p300 and GCN5. The addition of ATP enabled the acetyl-CoA to become accessible to these HATs, leading to efficient histone acetylation [4]. This experiment validates the notion that a local increase in ATP could make the sequestrated acetyl-CoA available to a consuming enzyme, which could be a HAT or ACC1 as described in a previous chapter.

## 6. NME1/2-Mediated Control of Gene Expression

A detailed consideration of NME2-mediated control of gene expression under the HFD showed that a change in histone H3K9 acetylation per se cannot account for the observed change in gene expression. Indeed, we have observed that in the absence of NME2, a category of genes escapes the physiological gene repression and another group resists activation. More specifically, we could demonstrate that a critical lipogenic transcription factor, SREBP1, escapes the NME2-dependent repression under the HFD. A continued activity of this transcription factor under HFD by its own can explain the maintained DNL under this condition. Indeed, its encoding gene, Srebp1c, as well as Acc and Acly, the master regulators of DNL, are all direct targets of SREBP, which binds their regulatory regions (including its own encoding gene, Srebp1c) and activates their expression [26]. Additionally, SREBP1 activates the SCAP encoding gene, which is involved in the processing and activation of SREBP1 [26]. Therefore, a decrease in SREBP1 during HFD feeding can by itself explain the inhibition of DNL. Now the important question is, how does NME2 control the activity of SREBP1?

SREBP1 is associated with the endoplasmic reticulum membrane as an inactive precursor protein, which needs to be cleaved by the SREBP cleavage-activation protein (SCAP). Additionally, the acetylation of SREBP1 at lysines 289 and 309 are critical for enhancing the transcription factor activity of this protein [26].

Therefore, we can speculate that the sequestration of acetyl-CoA by NME1/2 and its targeted release around specific chromatin regions restricts its usage for protein acetylation and hence prevents the acetylation-dependent activation of the SREBP1 transcription factor. The absence of NME2 leading to an increased availability of acetyl-CoA to CBP/p300 would maintain the acetylation of SREBP1, ensuring a continued activation of critical SREBP1 target genes involved in DNL, including Srebp1c itself.

## 7. Critical Evaluation of the Function of CoA/SCA-CoA Binding by NME1/2

The interaction of NME1 with CoA was first published by the Gout laboratory, who found that the binding of CoA by NME1 could lead to CoAlation of NME1 under oxidative stress [5]. These authors also show that CoAlated NME1 loses its NDPK activity. Therefore, this mechanism was proposed to be part of the cellular response to metabolic stress, particularly oxidative stress [5]. Our structural and biochemical investigations showed that NME1 binds CoA and ATP in a mutually exclusive manner, and that increasing concentration of CoA could inhibit NME1/2 catalytic histidine phosphorylation and hence its NDPK activity [4]. Therefore, not only does oxidative stress mediate NME1 CoAlation, but a simple non-covalent CoA binding by NME1/2 is also enough to inhibit the NDPK activity of these enzymes. Surprisingly, the third group, which reported CoA/acyl-CoA binding by NME1/2, also noted that NME1/2 NDPK activity could only be inhibited by long-chain fatty acyl-CoA and not by CoA or SCA-CoAs [6]. This result cannot be explained by the data discussed above and is even at odds with the authors’ own structural data, which report the crystal structure of NME2 with myristoyl-CoA or nonhydrolyzable myristoyl-CoA. Indeed, this study showed that the CoA moiety of these molecules occupies the active site where NDPs bind and that there is no contribution of the acyl moiety in this binding, which is in perfect agreement with two other published works on NME1-CoA structure [4,7]. Hence, the assertion made by the authors that CoA or SCA-CoAs cannot inhibit the NDPK activity of NME1/2 lacks structural clarity.

## 8. Concluding Remarks

The discovery of CoA/SCA-CoA binding activity of NME1/2 paves the way for considering a completely new cross-talk between two evolutionary ancient and critical groups of metabolites: NTP/NDPs and CoA/SCA-CoAs. Although the involvement of NME1/2 NDPK activity has been very deeply investigated and its regulatory impact on many cellular functions is established, the CoA/SCA-CoA-dependent role of NME1/2 is only emerging. Our work established the first insight into the new functional aspects of NME1/2 in the regulation of acetyl-CoA availability and use. However, important questions remain with regard to the interplay between NME1/2 NDPK activity and CoA/SCA-CoAs binding. Our data suggest that NME1/2 could be considered as both carriers of CoA/SCA-CoAs and enzymes involved in cellular nucleotide homeostasis, and that the local concentrations of NTPs and CoA/SCA-CoAs would determine the function of these enzymes. This new level of regulation based on an interplay between NTP/NDP and CoA/SCA-CoA could also involve other NME members with more marked intercellular localization, such as NME3 and NME4, which have also been found in the CoA-bound protein fraction [4]. Additionally, it is obvious that the regulatory mechanisms based on CoA/AC-CoA binding by NME1/2 are susceptible to also modulate classical functions of these proteins, based on their NDPK activity such as endocytosis and vesicle trafficking [2] (Figure 3).

## Figures and Tables

**Figure 1 ijms-25-07528-f001:**
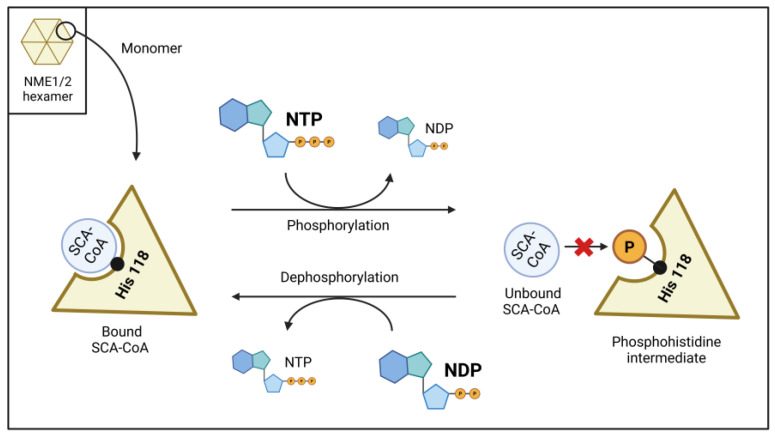
[NTP]/[NDP] Regulation of short-chain acyl-CoA (SCA-CoA) binding by NME1/2. NTP (physiologically mainly ATP) phosphorylates Histidine 118, effectively preventing SCA-CoA binding to NME1/2 until the phosphate group is transferred to an NDP molecule. In contrast, NDP promotes NME dephosphorylation by removing the phosphate from Histidine 118, thereby facilitating SCA-CoA binding. Higher levels of local NTP maintain NME1/2 in a predominantly phosphorylated state and unable to bind SCA-CoA, while elevated levels of local NDP promote NME dephosphorylation, enabling SCA-CoA binding.

**Figure 2 ijms-25-07528-f002:**
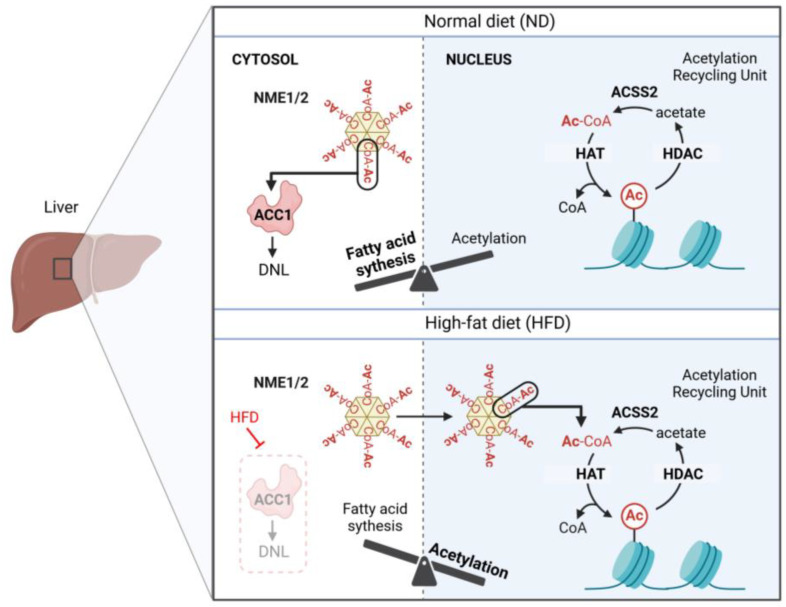
A hypothetical view of the control of competitive acetyl-CoA utilization in de novo lipogenesis (DNL) and histone acetylation by NME1/2. Under normal diet (ND), DNL is an active process in hepatocytes. The autonomous recycling of acetyl-CoA production and consumption in the nucleus makes the acetylation of histone independent of the cytosolic acetyl-CoA production, which is mostly used in DNL.Under these conditions, NME1/2 restricts the availability of acetyl-CoA to feed DNL (ACC1). Under HFD, the inhibition of DNL makes the NME1/2-loaded acetyl-CoA available to HATs for histone acetylation. Under these conditions, the local concentration of ATP in the vicinity of chromatin determines the privileged sites of acetyl-CoA release and histone acetylation.

**Figure 3 ijms-25-07528-f003:**
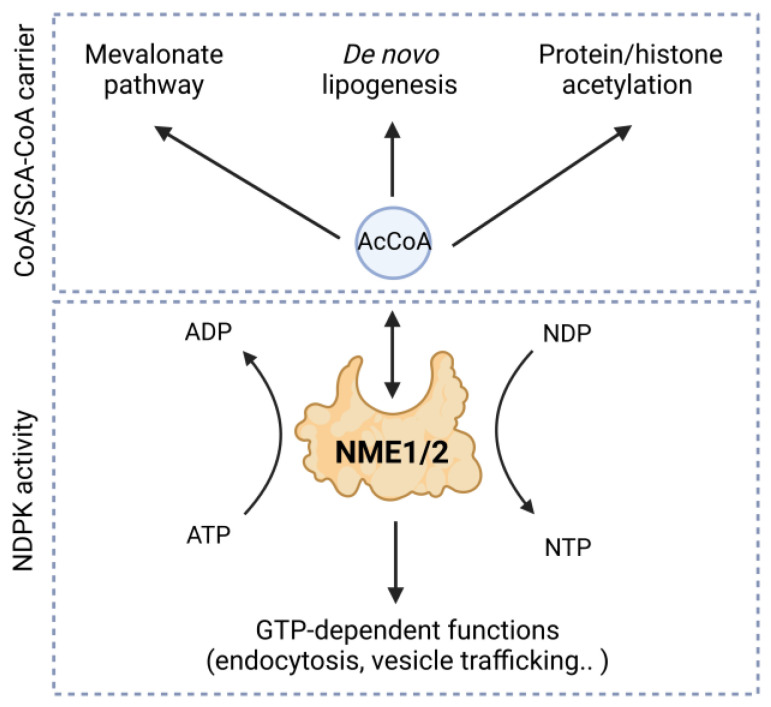
Illustration of cross-talk between two evolutionary ancient and critical groups of metabolites: NTP/NDPs and CoA/SCA-CoAs.

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
