# Peer review of "Nucleoside Diphosphate Kinases Are ATP-Regulated Carriers of Short-Chain Acyl-CoAs"

_ijms, 2024, doi:10.3390/ijms25147528_

Round 1

Reviewer 1 Report

Comments and Suggestions for Authors

This is a well-written short review covering a novel field in NME research. However, I would like the authors to address aome addtional issues.

1.) The authors attribute the binding of CoA/SCA-CoA to NME1 and NME2. Given the structural similaraties between Class 1 NMEs, is their any information on NME3 and NME4? If not, they should state so, but might want to discuss it.

2.) In this review, the authors regard NME1 and NME2 as equal, besides one exception, i.e. the regulation of histone acetylation, were a specific role of NME2 is reported. In the light of the recent data on heterooligomerization of different NME isoforms as a possibility to gain subcellular specificity, they should comment on heterohexamers and how the specificity might be achieved.

Minor point: As a reference to the important protein histidine kinase activity of NMEs, they cite a recent review of the group of Tony Hunter, which is accurate. Nevertheless, I would like to suggest to cite additionaly some original work, for example Cuello et al., JBC, 2003, in which the transfer of the high energetic phosphate bound to His118 of NME2 to a His residue in another protein, e. g. His266 in Gbeta, has been demonstrated for the first time.

Author Response

Thank you for your positive feedback on our review. We appreciate your suggestions, and we have addressed your comments below.

Comment 1.) The authors attribute the binding of CoA/SCA-CoA to NME1 and NME2. Given the structural similaraties between Class 1 NMEs, is their any information on NME3 and NME4? If not, they should state so, but might want to discuss it.

Response 1) Following the proteomics of spermatogenic cell CoA-binding factors published by Iuso and colleagues, in addition to NME1/2, NME3, and NME4 were also identified, indicating that they share a CoA-binding activity with NME1/2. However, quantitative considerations indicated that, compared with NME1/2, in the CoA-bound protein fraction they are present in much lower quantities. The reason for this could be their relative abundance compared with NME1/2 or their low extractability under the experimental conditions used. Indeed, NME3 (present on the cytosolic surface of mitochondria, the plasma membrane and potentially other membranes) could remain membrane-bound and NME4 (present in the mitochondrial matrix and intermembrane space) could remain masked to the CoA used for the pull-down experiment reported. The intracellular localizations of these proteins and their ability to bind CoA suggest restricted management of CoA/acetyl-CoA by NME3/4. NME3 may be involved in the management of cytosolic acyl-CoA close to membranes, while NME4 may be specifically involved in the management of mitochondrial acyl-CoA.

In the section 8 (concluding remarks), we have now included this information on the CoA-binding activity of NME3/4 (lines from 317 to 323). We've also mentioned the fact that, unlike NME1/2, NME3/4 would do so more restrictedly.

Comment 2.) In this review, the authors regard NME1 and NME2 as equal, besides one exception, i.e. the regulation of histone acetylation, were a specific role of NME2 is reported. In the light of the recent data on heterooligomerization of different NME isoforms as a possibility to gain subcellular specificity, they should comment on heterohexamers and how the specificity might be achieved.

Responce 2) In the manuscript by Iuso and colleagues, although in vivo physio-pathological studies involved the use of Nme2 ko mice, none of our biochemical analyses could show any specificity of NME1 or NME2 with regard to CoA binding and the competitive nature of NME1/2's ATP-dependent phosphorylation of catalytic histidine 118, and CoA binding by these proteins. Consequently, we have no argument in favor of a specific activity of NME1 compared with NME2 and thus in favor of a functional impact for heterohexamer formation.

Minor point) As a reference to the important protein histidine kinase activity of NMEs, they cite a recent review of the group of Tony Hunter, which is accurate. Nevertheless, I would like to suggest to cite additionaly some original work, for example Cuello et al., JBC, 2003, in which the transfer of the high energetic phosphate bound to His118 of NME2 to a His residue in another protein, e. g. His266 in Gbeta, has been demonstrated for the first time.

Response to Minor point: This reference in now cited, line 38, reference N°2

Thanks for your revision.

Best Regards

Reviewer 2 Report

Comments and Suggestions for Authors

This is an interesting and well-written review article, covering the recently detected ability of nucleoside diphosphate kinases (NME1/2) to bind CoA, acetyl-CoA, and acyl-CoA. This new aspect of NME function is covered here comprehensively, and several potential functional consequences are discussed systematically. However, a weakness of the article is the fact that the authors focus very much on these novel aspects, and in particular on their own paper in Science Advances. Thus, their conclusions appear in part a bit one-sided. Furthermore, the presented ideas are in part hypotheses and should be presented as such.

Specific comments:

By providing NTPs, NMEs control many different functions. In particular, both heterotrimeric and small G proteins are dependent on the supply of GTP, and this is important for many functions (e.g., endocytosis and vesicle trafficking). It would be an improvement of the article if the authors would not completely ignore these aspects.

The binding of CoA, acetyl-CoA and acyl-CoA to NMEs: sometimes the authors do not clearly differentiate here. For example, HAT enzymes are histone acetyl transferases, not acyl transferases (as stated on page 3 line 127). Is there really lysine acylation, or is it rather lysine acetylation (line 115)?In most cases, the authors talk about acetyl-CoA binding to NMEs. It would be helpful to make this point clearer.

CoA will very likely not require a carrier as it is highly water soluble (page 5). What is known about the cellular CoA levels, how are they regulated? Is there a competition between CoA and acetyl-CoA binding at the NMEs?

Figure 2 is not easy to understand. Isn`t there an equilibrium between free and NME-bound CoA-Ac also in the cytosol? The figure suggests that NME-bound CoA-Ac is available for ACC1 - other than explained in the text. Does HFD influence the levels of CoA-Ac and/or the expression of NME1/2? The figure furthermore suggests more histone acetylation in HFD, but the text explains that there is rather a re-localization of histone acetylation towards ATP-rich regions. Finally, HFD decreases de novo lipogenesis via transcriptional regulation of involved genes, which - if dependent on histone acetylation - would have a feedback effect on the availability of CoA-Ac. Thus, the availability of CoA-Ac would be both reason for changes in histone acetylation, and consequence thereof (because of suppressed de novo lipogenesis) in HFD. - The authors should make it clearer that these mechanisms are in large part hypothetical.

Minor points:

References 3 and 8 are identical.

When looking at the Figures, it is difficult to identify what is the caption, since it looks similar to the main body of the text.

Author Response

Thank you for your positive feedback on our review. We appreciate your suggestions, and we have addressed your comments below.

Comment 1) This is an interesting and well-written review article, covering the recently detected ability of nucleoside diphosphate kinases (NME1/2) to bind CoA, acetyl-CoA, and acyl-CoA. This new aspect of NME function is covered here comprehensively, and several potential functional consequences are discussed systematically. However, a weakness of the article is the fact that the authors focus very much on these novel aspects, and in particular on their own paper in Science Advances. Thus, their conclusions appear in part a bit one-sided. Furthermore, the presented ideas are in part hypotheses and should be presented as such.

Response 1) Among the three articles published on the NME1/2 CoA interaction, the level of functional studies is not equivalent. The article published in Science Advances presents the most comprehensive molecular studies. The other study using animal models (Zhang et al. Ref N°6) seems problematic because of an unexplained contradiction between the structural studies showing CoA binding by NME1/2 and their assertion that only long-chain acyl-CoA can bind and inhibit NME1/2 NDPK activity. This contradiction limits constructive discussion of these data and this point has been highlighted in our section 7 (Critical evaluation of the function of CoA/SCA-CoA binding by NME1/2).

Comment 2) By providing NTPs, NMEs control many different functions. In particular, both heterotrimeric and small G proteins are dependent on the supply of GTP, and this is important for many functions (e.g., endocytosis and vesicle trafficking). It would be an improvement of the article if the authors would not completely ignore these aspects.

Responce 2) We added a sentence to the discussion pointing out that CoA/Acetyl-CoA binding by NME1/2 could also impact cellular functions depending on their NTP-generating activity, such as endocytosis and cell trafficking. Please see section 8 (lines 322), concluding remarks and Fig.3.

Comment 3) The binding of CoA, acetyl-CoA and acyl-CoA to NMEs: sometimes the authors do not clearly differentiate here. For example, HAT enzymes are histone acetyl transferases, not acyl transferases (as stated on page 3 line 127). Is there really lysine acylation, or is it rather lysine acetylation (line 115)? In most cases, the authors talk about acetyl-CoA binding to NMEs. It would be helpful to make this point clearer.

Response 3) Acetyl-CoA is one of the short-chain acyl-CoAs (SCA-CoAs), and it has been shown that most of the major cellular HATs are also capable of utilizing SCA-CoAs in addition to acetyl-CoA. Furthermore, protein lysine can be similarly modified by acetylation or other types of acylations (butyrylation, crotonylation, etc.).

Comment 4) CoA will very likely not require a carrier as it is highly water soluble (page 5). What is known about the cellular CoA levels, how are they regulated? Is there a competition between CoA and acetyl-CoA binding at the NMEs?

Response 4) With regard to its solubility, CoA may not require a transporter, but our hypothesis is that NME1/2 could prevent the free diffusion of CoA and acyl-CoA and regulate their utilization according to cellular energy levels.

Comment 5) Figure 2 is not easy to understand. Isn`t there an equilibrium between free and NME-bound CoA-Ac also in the cytosol? The figure suggests that NME-bound CoA-Ac is available for ACC1 - other than explained in the text. Does HFD influence the levels of CoA-Ac and/or the expression of NME1/2? The figure furthermore suggests more histone acetylation in HFD, but the text explains that there is rather a re-localization of histone acetylation towards ATP-rich regions. Finally, HFD decreases de novo lipogenesis via transcriptional regulation of involved genes, which - if dependent on histone acetylation - would have a feedback effect on the availability of CoA-Ac. Thus, the availability of CoA-Ac would be both reason for changes in histone acetylation, and consequence thereof (because of suppressed de novo lipogenesis) in HFD. - The authors should make it clearer that these mechanisms are in large part hypothetical.

Response 5) ACSS2 has been shown to mediate a redistribution of histone acetylation in response to specific metabolic changes (see references 23 and 24). Our hypothesis is that in the absence of significant acetyl-CoA consumption in the cytoplasm, for example due to DNL inhibition, Ac-CoA-loaded NME1/2 could contribute to ACSS2-dependent redistribution of histone acetylation by supplying more acetyl-CoA to these same regions.

For Figure 2, we have modified the title, which now reads as follows: A hypothetical view of the control of competitive acetyl-CoA utilization in de novo lipogenesis (DNL) and histone acetylation by NME1/2.

Minor point 1) References 3 and 8 are identical.

Response to minor point 1) We are sorry for this error, which has been now corrected

Minor point 2) When looking at the Figures, it is difficult to identify what is the caption, since it looks similar to the main body of the text.

Response to minor point 2) We do not understand this remark.

Thanks for your revision.

Best regards.

Reviewer 3 Report

Comments and Suggestions for Authors

In this review, the authors summarized the major findings regarding the recently discovered functional activities of NME1/2 proteins and discussed their binding effect in the regulation of CoA and acyl-CoA. This review will definitely add a significant contribution to the field. However, the following points should be addressed before acceptance.

1. To make it clear to the reader, the authors should include in sections 4-7, figures illustrating different levels of control/binding rather than just long texts.

2. The authors suggested that NME1/2 could be used as storage container for acetyl-CoA for future use. How strong and reversible is the binding? Any available experimental data?

Comments on the Quality of English Language

English language is fine, minor editing is required.

Author Response

Thank you for your positive feedback on our review. We appreciate your suggestions, and we have addressed your comments below.

Comment 1) To make it clear to the reader, the authors should include in sections 4-7, figures illustrating different levels of control/binding rather than just long texts.

Response 1) We've added an additional Figure (Fig. 3) summarizing the section 8, illustrating new cross-talk between two evolutionary ancient and critical groups of metabolites: NTP/NDPs and CoA/SCA-CoAs

Comment 2) The authors suggested that NME1/2 could be used as storage container for acetyl-CoA for future use. How strong and reversible is the binding? Any available experimental data?

Response 2) Iuso and colleagues (Science Advances 2023) reported data on NME1 affinities for CoA in a range similar to or higher than for ADP (Kd for CoA of ~18 μM comparable to the Kd of 25 to 120 μM reported for the binding of NDPKs to ADP), and also calculated cell concentrations of NME 1/2 binding sites (~18 μM) and cytosolic CoA and acetyl-CoA (1-20 μM) to be within a similar order of magnitude.

However as discussed in the text, while the overall local concentrations of NTP/NDP could impact the competitive binding of CoA/AcCoA, the binding of NTPs not only competitively releases bound CoA/SCA-CoA but also phosphorylates His118, thereby stably preventing the rebinding of CoA/SCA-CoA. Conversely, NDP-dependent dephosphorylation of His118 promotes CoA/SCA-CoA binding. Consequently, high local NTP concentrations will favor the soluble pool of CoA/SCA-CoA, whereas high local NDP concentrations will favor NME1/2-bound CoA/SCA-CoA.

This question is therefore difficult to answer, as we are not dealing with a simple competitive binding between two types of metabolites, but also with a stable modification of the enzyme by NTP and its reversion by NDP.

Thanks for your revision,

Best Regards.